# Prevalence and Characteristics of Phenicol-Oxazolidinone Resistance Genes in *Enterococcus Faecalis* and *Enterococcus Faecium* Isolated from Food-Producing Animals and Meat in Korea

**DOI:** 10.3390/ijms222111335

**Published:** 2021-10-20

**Authors:** Eiseul Kim, So-Won Shin, Hyo-Sun Kwak, Min-Hyeok Cha, Seung-Min Yang, Yoon-Soo Gwak, Gun-Jo Woo, Hae-Yeong Kim

**Affiliations:** 1Institute of Life Sciences & Resources and Department of Food Science and Biotechnology, Kyung Hee University, Yongin 17104, Korea; eskim89@khu.ac.kr (E.K.); volume1@khu.ac.kr (S.-W.S.); hyosun610@gmail.com (H.-S.K.); ysm9284@gmail.com (S.-M.Y.); kys97112@naver.com (Y.-S.G.); 2Laboratory of Food Safety and Evaluation, Department of Biotechnology, Korea University Graduate School, Seoul 02841, Korea; cha900125@gmail.com

**Keywords:** *Enterococcus*, linezolid, antibiotic resistance gene, phenicol-oxazolidinone resistance gene, *optrA*, *fexA*, *poxtA*

## Abstract

The use of phenicol antibiotics in animals has increased. In recent years, it has been reported that the transferable gene mediates phenicol-oxazolidinone resistance. This study analyzed the prevalence and characteristics of phenicol-oxazolidinone resistance genes in *Enterococcus faecalis* and *Enterococcus faecium* isolated from food-producing animals and meat in Korea in 2018. Furthermore, for the first time, we reported the genome sequence of *E. faecalis* strain, which possesses the phenicol-oxazolidinone resistance gene on both the chromosome and plasmid. Among the 327 isolates, *optrA*, *poxtA*, and *fexA* genes were found in 15 (4.6%), 8 (2.5%), and 17 isolates (5.2%), respectively. Twenty *E. faecalis* strains carrying resistance genes belonged to eight sequence types (STs), and transferability was found in 17 isolates. The genome sequences revealed that resistant genes were present in the chromosome or plasmid, or both. In strains EFS17 and EFS108, *optrA* was located downstream of the *ermA* and *ant*(*9*)*-1* genes. The strains EFS36 and EFS108 harboring *poxtA*-encoding plasmid cocarried *fexA* and *cfr*(*D*). These islands also contained IS1216E or the transposon Tn554, enabling the horizontal transfer of the phenicol-oxazolidinone resistance with other antimicrobial-resistant genes. Our results suggest that it is necessary to promote the prudent use of antibiotics through continuous monitoring and reevaluation.

## 1. Introduction

Enterococci are symbiotic bacteria of the gastrointestinal tract of humans and animals [1]. However, enterococci infection can cause urinary tract and soft tissue infections, and in severe cases, it causes life-threatening infections such as septicemia and meningitis [2]. Enterococci of animal origin have the potential risk of transferring their antimicrobial resistance genes to human enterococci or disseminating to humans through the food chain [3]. Enterococci are also known as nosocomial pathogens, and *Enterococcus* species, especially *Enterococcus faecalis* and *E. faecium*, have become a particular clinical problem [1,4,5]. The use of florfenicol, phenicol antibiotics, has been increasing in animals to treat diseases caused by *E. faecalis* and *E. faecium* infections [5].

Linezolid, which belongs to oxazolidinone, is classified as a critically important category of antibiotics by the World Health Organization, meaning that it is a major antibiotic that requires prudent use [6]. Linezolid is the clinical last resort to treat drug-resistant Gram-positive bacteria, including vancomycin-resistant enterococci and methicillin-resistant *Staphylococcus aureus* (MRSA) [4,5,7]. The resistance to linezolid by Gram-positive bacteria can arise through two mechanisms [8]. The first is associated with a point mutation, such as G2576T or G2505A in the 23S ribosomal ribonucleic acid (rRNA) binding site, or mutations in the genes encoding ribosomal proteins, such as L3 or L4, forming the bacterial 50S subunit [7,8,9]. The second mechanism involves the acquisition of transferable resistance genes, such as *optrA* and *poxtA*, encoding the adenosine triphosphate-binding cassette F, as well as *cfr*, which encodes a methyltransferase [8]. These genes are considered as multiple resistance genes; *optrA* confers resistance to phenicols (chloramphenicol and florfenicol) and oxazolidinones (linezolid and tedizolid), *poxtA* confers resistance to phenicol-oxazolidinone-tetracycline, and *cfr* confers resistance to phenicols, oxazolidinones, lincosamides, streptogramin A, and pleuromutilins [8]. *optrA*, *poxtA*, and *cfr* genes present as part of a plasmid or as a transposon composite; it is known that the possibility of transferring these genes to other bacteria is very high [5].

Recently, linezolid-resistant enterococci and staphylococci have been reported in patients and also in food animals. Although the distribution rate of linezolid-resistant strains is very low, recent reports on the transferable *optrA* gene are increasing, and multidrug-resistant linezolid-resistant bacteria are emerging [9,10]. The *optrA*, *poxtA*, and *cfr* encoding enterococci has been reported in Korea and many countries such as China, Italy, Ireland, and Malaysia [3]. In particular, it was confirmed that all enterococci isolated from tertiary hospital patients in Korea possessed the *optrA* gene, but none of them were treated with linezolid [11]. Therefore, it was estimated that the *optrA*-positive linezolid nonsusceptible enterococci appeared at the hospital through the community onset [11]. The *optrA* gene has been continuously found in enterococci from foods of animal origin in Korea, and *optrA*-encoding enterococci cocarrying the phenicol exporter gene *fexA* have been isolated [3]. These genes are commonly embedded in mobile genetic elements as part of plasmids or present as composite transposons in the bacterial chromosome [5]. These characteristics of mobile genetic elements enable the rapid distribution of *optrA* with *poxtA* and *cfr* to the bacterial population. Therefore, it is important to understand the distribution of these antibiotic resistance genes.

In Korea, the use of phenicol antibiotics has steadily increased. It is reported that the amount of phenicols used as a veterinary medicinal product in 2019 is 110 tons, which is roughly twice that of 2010 [12]. This suggests that linezolid resistance is likely to increase along with the resistance to florfenicol. Florfenicol is a veterinary medicine used to control respiratory tract infections in livestock. However, since excessive use of florfenicol in animals can coselect the expression of phenicol-oxazolidinone resistance, the acquisition of cross-resistance between linezolid and florfenicol can potentially impact both humans and livestock [10]. Since this can significantly limit the treatment of multidrug-resistant bacteria, continuous monitoring and surveillance for the resistant bacteria and genes are required.

This study investigated the prevalence, antimicrobial resistance profile, and mobilizable nature of the phenicol-oxazolidinone resistance genes in *E. faecalis* and *E. faecium* isolated from food-producing animals and meat.

## 2. Results

### 2.1. Identification of Phenicol-Oxazolidinone Resistant Gene

Enterococci harboring the phenicol-oxazolidinone resistant genes *optrA*, *poxtA*, *cfr*, and *fexA* were detected from 282 *E. faecalis* and 45 *E. faecium* isolates. Among 282 *E. faecalis* isolates, the *optrA*, *poxtA*, and *fexA* genes were identified in 15 isolates (5.3%), six isolates (2.1%), and 17 isolates (6.0%), respectively, but the *cfr* gene was not detected. In 45 *E. faecium*, only two strains (4.4%) contained the *poxtA* gene, and no other genes were identified. As shown in Table 1, the phenicol-oxazolidinone resistance genes were detected in a total of 20 *E. faecalis* and two *E. faecium* isolates originating mostly from cattle (13/22), in addition to pigs (4/22) and fresh meat products including beef (2/22) and pork (3/22). Enterococci harboring multiple phenicol-oxazolidinone resistance genes were identified. Thirteen, four, and one *E*. *faecalis* isolates were shown to carry *optrA* and *fexA*, *poxtA* and *fexA*, and *optrA*, *poxtA,* and *fexA*, respectively. On the other hand, one *E. faecalis* had *optrA* only, and two of each *E*. *faecium* and *E*. *faecalis* isolates harbored *poxtA* only.

### 2.2. Antimicrobial-Resistant Pattern

Twenty-two enterococcal strains with phenicol-oxazolidinone resistance genes showed the highest resistance to quinupristin/dalfopristin (100%), florfenicol (100%), tetracycline (95%), erythromycin (95%), tylosin (95%), streptomycin (77%), chloramphenicol (77%), and kanamycin (64%), while the antimicrobial resistance level was low against gentamicin (14%), ciprofloxacin (9%), daptomycin (9%), tigecycline (5%), and ampicillin (5%). Linezolid-resistant enterococci were identified in three isolates (14%), and no vancomycin or salinomycin resistance was detected. Except for one strain, 21 strains (19 *E. faecalis* and two *E. faecium*) showed a multidrug resistance pattern (Table 2).

The minimum inhibitory concentration (MIC) of three linezolid-resistant strains was confirmed to be 8 mg/L, and the MIC of five linezolid-intermediate strains was determined to be 4 mg/L (Table 2). Among the florfenicol-resistant strains, a breakpoint of 16 mg/L or higher was confirmed in 21 strains, and of these, seven strains had a high MIC of 64 mg/L.

### 2.3. Molecular Typing by Multilocus Sequence Typing (MLST)

As shown in Table 3, the MLST analysis of 20 *E*. *faecalis* isolates revealed eight sequence types (STs) including ST593, ST100, ST16, ST585, ST915, ST338, ST47, and ST27. ST593 (clonal complex (CC) singleton) was the most predominant with nine strains (43%), and ST100 (CC100) and ST16 (CC16) were determined from three strains each, whilst ST585 (CC4), ST 915 (CC915), ST338 (CC4), ST47 (CC47), and ST27 (CC27) were each identified from a strain. The goeBURST analysis grouped all available STs into six CC and identified ST593, the most prevalent ST type in this study, as a singleton ST, as shown in the minimum spanning tree in Figure 1.

### 2.4. Transferability of Phenicol-Oxazolidinone Resistance Genes

The transferability of *optrA, poxtA,* and *fexA* genes was tested by the broth-mating method. It was confirmed that the resistance gene was transferred in 17 out of 22 (77%) enterococcal isolates. In two *E. faecium* strains, gene transfer was not identified. As shown in Table 1, each of the *optrA, poxtA*, and *fexA* genes possessed by each donor strain was successfully delivered to the recipient strain.

### 2.5. Complete Genome Sequencing

#### 2.5.1. Genome Features of Three Isolates

Complete genome sequencing was performed to identify the underlying antimicrobial resistance mechanisms. Genome analysis showed that the EFS17 strain consists of one chromosome and one plasmid, and EFS36 and EFS108 strains had two and five plasmids, respectively (Table 4). The genomic sizes of the strains EFS17, EFS36, and EFS108 consisted of 2,914,315 bp, 3,095,754 bp, and 3,093,936 bp with a GC content of 37.4%, 37.3%, and 37.3%, respectively. The genomic features of the EFS17 strain included 2803 coding genes and 73 coding regions for RNAs, of which 61 were transfer RNAs (tRNAs). Meanwhile, the EFS36 genome contained 3057 coding sequences and 78 coding regions for RNAs, including 66 tRNAs. The EFS108 genome displayed 2980 coding genes and 73 coding regions for RNAs, of which 61 were tRNAs.

#### 2.5.2. In Silico Identification of Antimicrobial Resistance Genes

A blastn search for known resistance genes revealed that three strains possessed the phenicol-oxazolidinone resistance *optrA* gene or *poxtA* gene; two strains also carried the *cfr*(*D*) gene (Figure 2). EFS17 and EFS108 strains harbored the *optrA* gene, whereas EFS36 and EFS108 strains had the *cfr*(*D*) and *poxtA* genes. The phenicol-oxazolidinone resistance genes were present on the chromosome in the EFS17 strain; for the EFS36 strain, this gene was present on the plasmid (pEFS36_2). In the EFS108 strain, phenicol-oxazolidinone resistance genes were present in both the chromosome and plasmid (pEFS108_1) (Figure 2).

The EFS17 strain harbored *optrA* and *fexA* phenicol resistance genes. In the EFS17 strain, the *fexA* gene was encoded 689 bp upstream of the *optrA* gene, whereas the EFS108 strain lacked *fexA* (Appendix A). Transposon Tn554 was detected downstream of *optrA* in EFS17 and EFS108 strains (Figure 2). Moreover, the *ermA* and *ant*(*9*)-*I* genes conferring resistance to erythromycin and spectinomycin antibiotic were located upstream of the *optrA* genes in both EFS17 and EFS108 isolates. Additionally, analysis of complete genome sequence revealed that the EFS17 strain included additional genes for resistance to macrolide-lincosamide-streptogramin B (*ermA*, *ermB*, *lunB*, *lasA*, and *lasE*), aminoglycosides (*aac*(*6*′)-*aph*(*2*″), *aph*(*3*′)-*III*, *str*, and *ant*(*6*)*-Ia*), and tetracyclines (*tetM* and *tetL*) (Table 5). EFS36 and EFS108 strains harbored a *poxtA*-carrying plasmid named pEFS36_2 and pEFS108_1, respectively (Figure 3). Plasmid pEFS36_2 was 35,757 bp in length with a GC content of 34.3%. Plasmid pEFS108_1 was 97,455 bp in length with a GC content of 33.8%. These plasmids coharbor *fexA*, *cfr(D)*, and IS1216E, which were identified in two or three places. Therefore, EFS17, EFS36, and EFS108 contained several mobile gene elements and transposase-associated genes, enabling the horizontal transfer of the phenicol-oxazolidinone resistance genes.

## 3. Discussion

Knowledge of the distribution of antimicrobial-resistant strains in the food chain and food animals is important in determining the potential risk to human health [13]. Here, we investigated the prevalence and genetic characterization of phenicol-oxazolidinone resistance genes to understand better the phenicol-oxazolidinone resistance profiles of enterococci isolates obtained from food animals, animal carcass, and meat in Korea. Moreover, for the first time in Korea, this study reports the complete genome sequence of the *E. faecalis* EFS108 strain harboring phenicol-oxazolidinone resistant genes on both the chromosome and the plasmid.

Overall, 6.7% (22/327) of the enterococci isolates tested harbored phenicol-oxazolidinone resistance genes. Among them, *optrA, poxtA,* and *fexA* were detected in 15 isolates (4.6%), 8 isolates (2.5%), and 17 isolates (5.2%), respectively, confirming that these genes were distributed at a low level. Consistent with this study, a low phenicol-oxazolidinone resistance rate was reported in *E. faecalis* and *E. faecium* from food animals and animal carcasses in Korea [1,3,14,15], China [16], and Europe [17]. In a previous study, enterococci strains with phenicol-oxazolidinone resistance genes were isolated at slightly higher rates in *E. faecalis* than *E. faecium*, and the *optrA* gene was confirmed only in *E. faecalis* isolates [18]. Our results are congruent with reports from previous studies where *optrA* is predominantly present in *E. faecalis*, and the reason is presumed that the loss of conjugated resistance plasmids occurs in *E. faecium* [4,19]. Moreover, none of the phenicol-oxazolidinone resistant enterococci strains carried the *cfr* gene, but the *cfr*(*D*) gene was confirmed through complete genome sequencing in two *E. faecalis* isolates (EFS36 and EFS108). Therefore, when monitoring the phenicol-oxazolidinone resistance gene, it will be necessary to examine the *cfr* gene and its variants. The reason is that horizontal gene transfer is possible between *cfr* gene-carrying strains. In addition to the *cfr* gene, variants *cfr*(*B*), *cfr*(*C*), and *cfr*(*D*) have been reported in clinical isolates [9,10].

Twenty-two enterococcal isolates harboring phenicol-oxazolidinone resistance genes were resistant to one or more antimicrobial agents. Multidrug-resistant isolates were observed in both *E. faecalis* (95%) and *E. faecium* (100%) with high frequencies. A high resistance rate was observed for quinupristin/dalfopristin, florfenicol, tetracycline, erythromycin, and tylosin. Antimicrobial patterns were similar to those of other food animals. The high-level resistance to phenicols, tetracycline, and macrolides (erythromycin and tylosin) was also found in food animals and their carcass in Korea [1] and European countries [20]. These antimicrobial agents are generally administered to food animals in Korea [1]. Resistance to linezolid was observed in three *E. faecalis* isolates but not in *E. faecium*. All linezolid-resistant *E. faecalis* isolates were also resistant to florfenicol and chloramphenicol.

In the MLST analysis to investigate the clonal relationship of phenicol-oxazolidinone resistance strains, the STs of enterococci isolated from meat, slaughterhouses, and farms could be classified into eight different types. Consistent with this study, the diversity of STs was also reported in enterococci isolated from food animals and animal carcass in Korea [3,8] and humans and animals in China [21]. Some *E. faecalis* strains isolated from different samples had *optrA* and *fexA* genes commonly, and the ST was the same as 593. From this result, resistance could be acquired through antibiotics for animals such as florfenicol in the farm. It can be estimated that resistance genes *optrA* and *fexA* may be spread to animals within the farm through feed or the farm environment [1]. ST16 was identified in three *optrA*-positive *E. faecalis*; two isolates cocarried *fexA,* and one isolate has only *optrA*. This ST has been identified in food animals and humans in Korea and other countries and is known as an *optrA*-carrying clone with a worldwide distribution [11,19]. In addition, EFS17 isolated from pork was identified as ST585, which coharbored *optrA* and *fexA*. ST585 *E. faecalis* harboring *optrA* has also been reported in the United States, China, and Germany [11].

Complete genome sequencing was performed on three *E. faecalis* isolates (EFS17, EFS36, and EFS108), which have different combinations of phenicol resistance genes, to investigate the location and genetic environment of *optrA* and *poxtA*. According to a report investigating the genetic environment of *optrA*-carrying plasmid in *E. faecalis* originated from humans and animals, the IS1216E element was detected either upstream or downstream of the *optrA* gene [22]. In addition, *fexA* and *erm*(*A*)-related genes were found upstream and downstream of *optrA*, respectively [22]. Conversely, when *optrA* is located in the chromosome, the *optrA* gene was located immediately downstream of the transcriptional regulator gene *araC* or downstream of the *fexA* gene [22]. Alternatively, the integration of the *optrA* gene was observed between Tn558 and Tn554 relics located in the chromosomal *radC* gene [22]. The *radC* gene is the common integration site for transposons of the Tn554 family [22]. Our study found that the *optrA* gene was located on the EFS17 and EFS108 genomes but not in the EFS36 genome. As in a previous study [15], the *optrA* gene on the EFS108 chromosome was identified downstream of *araC*, whereas the *optrA* gene on the EFS17 chromosome was found downstream of *fexA*. Moreover, the chromosomal *radC* gene was found in both EFS17 and EFS108.

Phenicol-oxazolidinone resistance has been reported to be mediated by the transferable phenicol-oxazolidinone resistance gene in recent years [1,16]. The *optrA* gene in strains EFS36 and EFS108 was located adjacent to Tn554 transposon. Tn554 was also found upstream of the *optrA* gene in other *Enterococcus* strains and *Staphylococcus sciuri* [23]. The functionally active Tn554 can be excised from their host DNA and generate circular forms that precede the integration of the transposon into a target sequence [23,24]. A similar genetic arrangement of *optrA* and Tn554 was identified in enterococci and staphylococci [13,16], suggesting that *optrA* can be mediated for transfer between different bacterial genus and species. The *optrA* gene was flanked by transposon or insertion sequences, indicating that a mobile genetic element mediates the horizontal transfer of *optrA* between bacteria of different genera, which should be given more attention to avoid disseminating the oxazolidinone resistance gene in the environment.

The *poxtA* gene was located on plasmids in two enterococci strains. The *poxtA* gene is often surrounded by bacteria insertion sequences on the plasmid in staphylococci or enterococci strains [13,16]. Our data showed that all *poxtA* found on plasmids were flanked by IS1216E, similar to the previous studies [9,16,25]. IS1216E belongs to the IS6 family of the bacteria insertion sequence, which among other things mediates transmission of the macrolide-lincosamide-streptogramin B resistance genes *erm*(*B*) in *E. hirae*, the oxazolidinone resistance gene *cfr* in *E. faecalis*, vancomycin resistance gene *vanA* in *E. faecium*, and tetracycline resistance gene *tetS* in *Streptococcus infantis* [23,26,27,28]. This indicates that *poxtA* can be transferred between different bacterial genera by IS-mediated recombination events, which plays an important role in disseminating antimicrobial resistance genes. Furthermore, two plasmids (pEFS36_2 and pEFS108_1) carrying *poxtA* gene also coharbored other resistance genes, such as *fexA* and *cfr*(*D*). This result suggests that *poxtA* together with *fexA* and *cfr*(*D*) genes may be disseminated among different strains via IS1216E.

The transferability of phenicol-oxazolidinone resistance genes was confirmed through an experiment by broth mating and a genetic context analysis using complete genome sequencing. The transferable phenicol-oxazolidinone resistant genes, *optrA* and *poxtA,* were detected by the broth mating method in 12 and four *E. faecalis* isolates. All of them, except for two isolates harboring *poxtA*, cocarried the phenicol exporter gene *fexA*. These results were also confirmed by whole-genome sequencing analysis. Genome analysis showed that transposons or insertion sequences mediating resistance genes were adjacent to *optrA*, *poxtA*, and *fexA* genes.

In conclusion, although the occurrence of a phenicol-oxazolidinone resistance gene in enterococci is still rare among food animals, a high rate of transferable phenicol-oxazolidinone genes was observed in these strains. The emergence of these genes in enterococci isolates from food animals is a serious problem, as they can be transmitted to humans through the food chain. The spread of these genes can significantly limit the treatment against multidrug-resistant bacteria. Hence, active surveillance of phenicol-oxazolidinone-resistant bacteria and related resistance genes is essential to prevent the spread of resistant enterococci isolates.

## 4. Materials and Methods

### 4.1. Enterococcus Strains

In 2018, 282 *E. faecalis* strains and 45 *E. faecium* strains were isolated in Korea from 43 meat (19 beef and 24 pork), 24 slaughterhouses, and 16 farms. In detail, 128 *E. faecalis* and 9 *E. faecium* strains were isolated from meat, and 48 *E. faecalis* and 13 *E. faecium* strains were isolated from the slaughterhouse. The remaining 106 *E. faecalis* and 23 *E. faecium* strains were isolated from the farm. All isolates were used in the experiment after being identified as enterococci using VITEK^®^ MS (bioMérieux, Marcy l’Etoile, France). All enterococci strains were cultured on tryptone soya agar (TSA) medium at 37 °C for 24 h.

### 4.2. Screening of Phenicol-Oxazolidinone Resistance Gene by Polymerase Chain Reaction (PCR)

Genomic DNA from *Enterococcus* isolates was extracted using the DNeasy Blood & Tissue kit (Qiagen, Hilden, Germany) according to the manufacturer’s instructions. The presence or absence of the phenicol-oxazolidinone resistance genes, such as *optrA, poxtA, cfr*, and *fexA*, in the isolates was detected by PCR according to the previous studies [9,24]. The primer sequences of each gene are shown in Table 6. The PCR mixture (25 µL) contained 1 µL of each primer (0.4 µM), 0.1 µL of *Taq* polymerase (5 U/µL), 2.5 µL of 10× buffer, 16.4 µL of distilled water, and 2 µL of template DNA. Amplification for the *optrA*, *poxtA*, and *cfr* genes was performed at 95 °C for 2 min, and 25 cycles of 95 °C for 15 s, 53 °C for 15 s, 68 °C for 1 min, and final elongation at 68 °C for 5 min. PCR conditions for the *fexA* gene were as follows: initial denaturation at 94 °C for 5 min, 35 cycles at 94 °C for 1 min, 57 °C for 1 min, 72 °C for 1.5 min, and final elongation at 72 °C for 5 min. The PCR-positive control DNA for *optrA*, *poxtA*, *cfr*, and *fexA* was kindly provided from the Animal and Plant Quarantine Agency in Korea.

### 4.3. Antimicrobial Susceptibility Testing

An antimicrobial susceptibility test was performed on the 22 *E. faecalis* and *E. faecium* isolates harboring at least one of the phenicol-oxazolidinone resistance genes *optrA*, *poxtA, cfr*, and *fexA*. The MIC of 16 antimicrobial agents was determined by the broth microdilution method using commercially available Sensititre1 panel KRVP2F (TREK Diagnostic Systems, West Sussex, UK), according to the manufacturer’s instructions. *Staphylococcus aureus* ATCC 29213 and *E. faecalis* ATCC 29212 were used as the quality control strains. The interpretation of the results followed the Clinical and Laboratory Standards Institute guidelines [29]. Sixteen antibiotics used in this study were as follows: gentamycin (GEN), streptomycin (STR), kanamycin (KAM), ampicillin (AMP), ciprofloxacin (CIP), vancomycin (VAN), tigecycline (TGC), erythromycin (ERY), tylosin (TYLT), linezolid (LZD), chloramphenicol (CHL), florfenicol (FFN), quinupristin/dalfopristin (SYN), tetracycline (TET), daptomycin (DAP), and salinomycin (SAL).

### 4.4. MLST Analysis

MLST analysis of 20 *E. faecalis* isolates harboring phenicol-oxazolidinone resistance genes was performed according to a previous study [30]. Seven housekeeping genes, such as *gdh*, *gyd*, *pstS*, *gki*, *aroE*, *xpt*, and *yqiL*, were amplified by PCR and then sequenced. Sequence types of *E. faecalis* were determined using the *E. faecalis* MLST database in PubMLST (http://pubmlst.org/efaecalis/, accessed on 22 September 2021). The clonal relationship between STs was analyzed by the goeBURST algorithm using the PHYLOViZ software v 2.0 [31].

### 4.5. Conjugation

Twenty-two *E. faecalis* and *E. faecium* isolates harboring phenicol-oxazolidinone resistance genes were used as the donor cells, and with *Escherichia coli* strain J53 served as the recipient strain for the conjugation experiments using methods previously described by Aljahdali et al. with minor modifications [32]. Briefly, each donor and recipient cells were cultured on TSA agar and incubated at 37 °C for 24 h. Cultured donor and recipient cells were mixed in 500 µL LB broth (1:1) and incubated at 37 °C for 3 h. Transconjugants were streaked on LB agar containing 200 mg/L sodium azide and incubated at 37 °C for 24 h. Several colonies were picked and plated on MacConkey agar at 37 °C for 24 h. Transconjugants from MacConkey agar were subcultured on TSA, incubated at 37 °C for 24 h, and subjected to PCR for the *optrA*, *poxtA*, *cfr*, and *fexA* genes, respectively, according to Table 6.

### 4.6. Complete Genome Sequencing

#### 4.6.1. Genome Sequencing, Assembly, and Annotation

Total genomic DNAs from three strains were extracted using the DNeasy Blood and Tissue kit (Qiagen, Hilden, Germany) following the manufacturer’s instructions. Libraries were prepared using the SMRTbell Express Template Prep kit 2.0 (Pacific Biosciences, Menlo Park, CA, USA) and subsequently sequenced using PacBio Sequel Ⅱe (Pacific Biosciences). Raw reads were processed using demultiplex barcodes protocol, which is part of SMRT Link version 10.1, to obtain clean reads. Clean data were assembled using the microbial assembly protocol in the SMRT Link software. Genome annotations were performed using Rapid Annotation using Subsystem Technology server version 2.0 with default parameters [33]. The complete genome sequences of *E*. *faecalis* EFS17, EFS36, and EFS108 were deposited in GenBank under the accession numbers SUB10526591, SUB10526592, and SUB10526593, respectively.

#### 4.6.2. Bioinformatics Analysis

The plasmid was identified using PlasmidFinder version 2.0 [34]. The identification of antimicrobial resistance genes in three genomes was performed using ResFinder version 4.1 [35]. ResFinder was set with default parameters, 90% identity threshold, and 60% minimum length threshold. The circular plasmid maps of pEFS36_2 and pEFS108_1 were generated using SnapGene Viewer (https://www.snapgene.com/snapgene-viewer/, accessed on 10 September 2021).

## Figures and Tables

**Figure 1 ijms-22-11335-f001:**
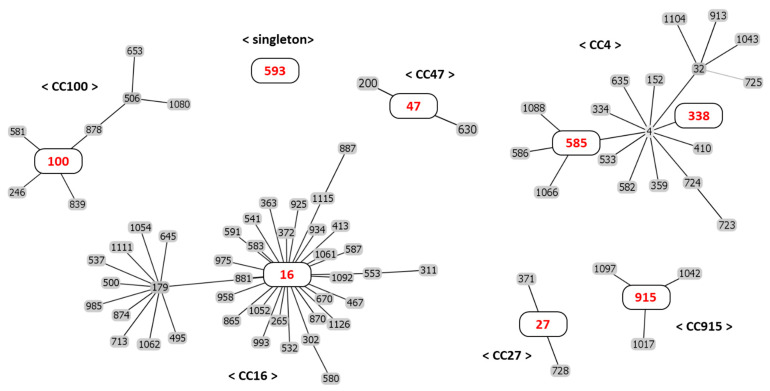
Graphical presentation of multilocus sequence typing data. Each ST analyzed in this study marked in red. The lines connect single-locus variants. The clonal relatedness between sequence types (STs) was analyzed against the entire *E. faecalis* database. Clonal complexes (CCs) are indicated. Eight sequence types (STs) were identified among the 20 *E. faecalis* isolates: ST593, ST100, ST16, ST27, ST338, ST47, ST585, and ST915. The eBURST algorithm clustered the STs into one singleton and six clonal complexes: CC100, CC16, CC27, CC4, CC47, and CC915.

**Figure 2 ijms-22-11335-f002:**
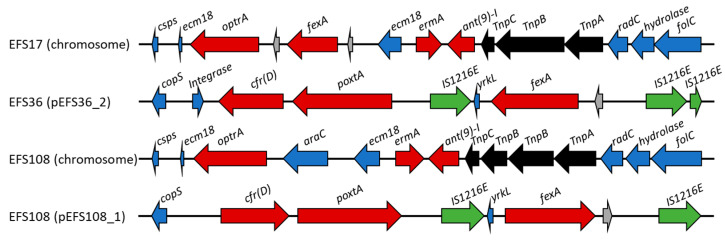
Schematic representation of phenicol-oxazolidinone resistance genes loci in three *E. faecalis* genomes. Genes and their orientation are shown with arrows and labeled; red, green, black, blue, and gray indicate antibiotic resistance gene, insertion sequences (ISs), transposase, known proteins, and hypothetical proteins, respectively. Csps, cold shock protein of CSP family; *ecm18*, class I SAM-dependent methyltransferase; *radC*, DNA repair protein RadC; *folC*, dihydrofolate synthase; *yrkL*, NAD(P)H oxidoreductase YRKL.

**Figure 3 ijms-22-11335-f003:**
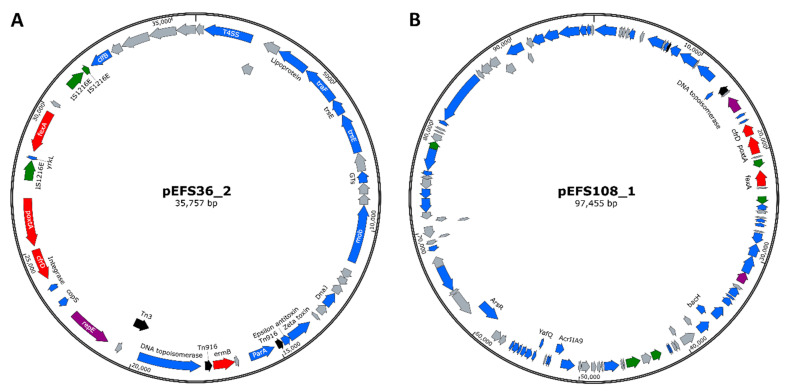
Schematic diagram of genetic structure of *poxtA* linezolid resistance gene-containing plasmids (**A**) pEFS36_2 from EFS36 isolate and (**B**) pEFS108_1 from EFS108 isolate. Genes and their orientation are indicated by arrow as follows: red, green, black, purple, blue, and gray represent antibiotic resistance gene, ISs, transposase, rep protein, other protein, and hypothetical proteins, respectively.

**Table 1 ijms-22-11335-t001:** Characteristics of enterococci strains harboring phenicol-oxazolidinone resistance genes.

Species	Strain	Source	Detected Resistance Genes by PCR				Transconjugants		
			*optrA*	*poxtA*	*cfr*	*fexA*	*optrA*	*poxtA*	*fexA*
*E. faecalis*	EFS17	Pork, meat	+	−	−	+	−	−	−
*E. faecalis*	EFS24	Pork, meat	+	−	−	+	+	−	+
*E. faecalis*	EFS27	Pork, meat	+	−	−	+	+	−	+
*E. faecalis*	EFS36	Beef, meat	−	+	−	+	−	+	+
*E. faecalis*	EFS74	Beef, meat	−	+	−	−	−	+	−
*E. faecalis*	EFS98	Pig, slaughterhouse	−	+	−	+	−	−	−
*E. faecalis*	EFS99	Pig, slaughterhouse	−	+	−	+	−	−	+
*E. faecalis*	EFS108	Pig, slaughterhouse	+	+	−	+	−	+	+
*E. faecalis*	EFS117	Cattle, slaughterhouse	+	−	−	−	−	−	−
*E. faecalis*	EFS147	Cattle, farm	+	−	−	+	+	−	+
*E. faecalis*	EFS151	Cattle, farm	+	−	−	+	+	−	+
*E. faecalis*	EFS153	Cattle, farm	+	−	−	+	+	−	+
*E. faecalis*	EFS154	Cattle, farm	+	−	−	+	+	−	+
*E. faecalis*	EFS158	Cattle, farm	+	−	−	+	+	−	+
*E. faecalis*	EFS253	Pig, slaughterhouse	+	−	−	+	+	−	+
*E. faecalis*	EFS255	Cattle, slaughterhouse	−	+	−	−	−	+	−
*E. faecalis*	EFS268	Cattle, farm	+	−	−	+	+	−	+
*E. faecalis*	EFS269	Cattle, farm	+	−	−	+	+	−	+
*E. faecalis*	EFS270	Cattle, farm	+	−	−	+	+	−	+
*E. faecalis*	EFS271	Cattle, farm	+	−	−	+	+	−	+
*E. faecium*	EFM21	Cattle, farm	−	+	−	−	−	−	−
*E. faecium*	EFM262	Cattle, farm	−	+	−	−	−	−	−

+, indicates the presence of phenicol-oxazolidinone resistance genes or transferability; −, indicates absence of resistance genes or not transferability.

**Table 2 ijms-22-11335-t002:** Antimicrobial resistance patterns of enterococci strains harboring phenicol-oxazolidinone resistance genes.

Scheme 1.	Strain	Antimicrobial Resistance Pattern ^1^	MIC ^2^ (mg/L)	
			LZD	FFN
*E*. *faecalis*	EFS17	TET, CIP, ERY, TYLT, LZD, GEN, KAN, STR, CHL, FFN	8	64
*E*. *faecalis*	EFS24	TET, ERY, TYLT, LZD, GEN, KAN, STR, CHL, FFN	8	64
*E*. *faecalis*	EFS27	TET, ERY, TYLT, KAN, STR, CHL, FFN	4	64
*E*. *faecalis*	EFS36	ERY, TYLT, GEN, KAN, STR, CHL, FFN	2	64
*E*. *faecalis*	EFS74	TET, FFN	4	16
*E*. *faecalis*	EFS98	TET, ERY, TYLT, STR, FFN	4	64
*E*. *faecalis*	EFS99	TET, ERY, TYLT, STR, FFN	4	16
*E*. *faecalis*	EFS108	TET, ERY, TYLT, CHL, FFN	2	64
*E*. *faecalis*	EFS117	TET, DAP, ERY, TYLT, FFN	−	16
*E*. *faecalis*	EFS147	TET, ERY, TYLT, CHL, FFN	−	32
*E*. *faecalis*	EFS151	TET, ERY, TYLT, KAN, STR, CHL, FFN	−	32
*E*. *faecalis*	EFS153	TET, ERY, TYLT, KAN, STR, CHL, FFN	−	32
*E*. *faecalis*	EFS154	TET, ERY, TYLT, STR, CHL, FFN	−	32
*E*. *faecalis*	EFS158	TET, ERY, TYLT, KAN, STR, CHL, FFN	−	32
*E*. *faecalis*	EFS253	TET, ERY, TYLT, LZD, GEN, KAN, STR, CHL, FFN	8	64
*E*. *faecalis*	EFS255	TET, ERY, TYLT, FFN	−	16
*E*. *faecalis*	EFS268	TET, ERY, TYLT, KAN, STR, CHL, FFN	−	32
*E*. *faecalis*	EFS269	TET, ERY, TYLT, KAN, STR, CHL, FFN	−	32
*E*. *faecalis*	EFS270	TET, ERY, TYLT, KAN, STR, CHL, FFN	−	32
*E*. *faecalis*	EFS271	TET, TGC, ERY, TYLT, KAN, STR, CHL, FFN	−	32
*E*. *faecium*	EFM21	TET, CIP, DAP, ERY, TYLT, KAN, STR, CHL, FFN	4	32
*E*. *faecium*	EFM262	TET, TGC, CIP, ERY, TYLT, KAN, STR, AMP, CHL, FFN	−	32

^1^ AMP, ampicillin; CHL, chloramphenicol; CIP, ciprofloxa-cin; DAP, daptomycin; ERY, erythromycin; FFN, florfenicol; GEN, gentamycin; KAN, kanamycin; LZD, linezolid; STR, streptomycin; TET, tetracycline; TGC, tigecycline; TYLT, tylosin; ^2^ MIC, minimum inhibitory concentration; LNZ, linezolid; FFN, florfenicol.

**Table 3 ijms-22-11335-t003:** MLST analysis of enterococci strains harboring phenicol-oxazolidinone resistance genes.

Strain	Allele							ST ^1^	CC ^2^
	*gdh*	*gyd*	*pstS*	*gki*	*aroE*	*xpt*	*yiqL*		
EFS17	8	7	7	4	4	4	1	585	4
EFS24	5	1	1	3	7	7	6	16	16
EFS27	39	2	49	45	7	2	17	915	915
EFS36	8	7	7	5	4	14	1	338	4
EFS74	19	1	24	22	19	17	14	47	47
EFS98	34	2	17	37	29	23	6	100	100
EFS99	34	2	17	37	29	23	6	100	100
EFS108	3	2	7	10	10	2	7	27	27
EFS117	5	1	1	3	7	7	6	16	16
EFS147	14	2	17	1	3	3	17	593	Singleton
EFS151	14	2	17	1	3	3	17	593	Singleton
EFS153	14	2	17	1	3	3	17	593	Singleton
EFS154	14	2	17	1	3	3	17	593	Singleton
EFS158	14	2	17	1	3	3	17	593	Singleton
EFS253	5	1	1	3	7	7	6	16	16
EFS255	34	2	17	37	29	23	6	100	100
EFS268	14	2	17	1	3	3	17	593	Singleton
EFS269	14	2	17	1	3	3	17	593	Singleton
EFS270	14	2	17	1	3	3	17	593	Singleton
EFS271	14	2	17	1	3	3	17	593	Singleton

^1^ ST, sequence type; ^2^ CC, clonal complex.

**Table 4 ijms-22-11335-t004:** Genomic features of three *E. faecalis* isolates.

Genomic Features	EFS17	EFS36	EFS108
Genome size (bp)	2,914,315	3,095,754	3,093,936
Chromosome size (bp)	2,838,954	3,016,592	2,816,588
Number of plasmids	1	2	5
Size range of plasmid(s) (bp)	75,361	35,757–43,405	5120–97,455
GC content (%)	37.4	37.3	37.3
Number of genes	2876	3135	3053
Number of coding genes	2803	3057	2980
Coding genes in chromosome	2712	2964	2670
Coding genes in plasmid(s)	91	93	310
Number of RNAs	73	78	73

**Table 5 ijms-22-11335-t005:** Antimicrobial resistance genes in three *E. faecalis* strains.

Strain	Class	Antimicrobial Resistance Genes
EFS17	aminocyclitol	*ant(9)-Ia*
	aminoglycoside	*aac*(*6*′)*-aph*(*2*″), *aph*(*3*′)-*III*, *str*, *ant*(*6*)*-Ia*
	fluoroquinolone	*parC*
	folate pathway antagonist	*dfrG*
	lincosamide	*lsaA*, *lsaE*, *lnuB*, *ermA*, *ermB*
	macrolide	*ermA*, *ermB*
	oxazolidinone	*optrA*
	phenicol	*fexA*, *optrA*, *cat*
	pleuromutilin	*lsaE*
	streptogramin a	*lsaA*, *lsaE*
	streptogramin b	*ermA*, *ermB*
	tetracycline	*tetM*, *tetL*
EFS36	aminoglycoside	*aac*(*6*′)*-aph*(*2*″), *aph*(*3*′)*-III*, *ant*(*6*)*-Ia*
	lincosamide	*lsaE*, *cfr*(*D*), *lnuB*, *lsaA*, *ermB*
	macrolide	*ermB*
	oxazolidinone	*cfr*(*D*), *poxtA*
	phenicol	*fexA*, *poxtA*, *cfr*(*D*)
	pleuromutilin	*lsaE*, *cfr*(*D*)
	streptogramin a	*lsaA*, *lsaE*, *cfr*(*D*)
	streptogramin b	*ermB*
	tetracycline	*poxtA*
EFS108	aminocyclitol	*ant*(*9*)*-Ia*
	*lincosamide*	*ermA*, *ermB*, *lsaA*
	*macrolide*	*ermA, ermB*
	*oxazolidinone*	*optrA, cfr*(*D*)*, poxtA*
	*phenicol*	*fexA*, *optrA*, *cat*
	*streptogramin a*	*lsaA*
	*streptogramin b*	*ermA*, *ermB*
	*tetracycline*	*tetM*, *tetL*

**Table 6 ijms-22-11335-t006:** Primer sequences for the detection of phenicol-oxazolidinone resistance genes.

Target	Primer	Nucleotide Sequence (5′→3′)	Amplicon (bp)	Reference
*cfr*	cfr-F	TGC TAC AGG CGA CAT TGG AT	137	[9]
	cfr-R	GAC GGT TGG CTA GAG CTT CA		
*optrA*	optrA-F	ACC GGT GTC CTC TTT GTC AG	369	[9]
	optrA-R	TCA ATG GAG TTA CGA TCG CCT T		
*poxtA*	poxtA-F	TCA GAG CCG TAC TGA GCA AC	167	[9]
	poxtA-R	CGT TTC TGG GTC AAG GTG GT		
*fexA*	fexA-F	GTA CTT GTA GGT GCA ATT ACG GCT GA	1272	[24]
	fexA-R	CGC ATC TGA GTA GGA CAT AGC GTC		

## Data Availability

The data presented in this study are available on request from the corresponding author.

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
