# Peer review of "Prevalence and Characteristics of Phenicol-Oxazolidinone Resistance Genes in Enterococcus Faecalis and Enterococcus Faecium Isolated from Food-Producing Animals and Meat in Korea"

_ijms, 2021, doi:10.3390/ijms222111335_

Round 1

Reviewer 1 Report

The manuscript entitled "Prevalence and characteristics of phenicol-oxazolidinone resistance genes in Enterococcus faecalis and Enterococcus faecium isolated from food-producing animals and meat in Korea" is generally well written, with clear presentations. I have only one primary question here and would also like to suggest some minor changes to this work. Please read below for further details..

In E. faecalis strain EFS108, the authors showed that the optrA gene is located on the chromosome, but not in the plasmid. However, the plasmid conjugation experiment demonstrated that this gene has successfully been acquired by the E. coli J53 recipient strain. How did it happen? Can the authors please explain?

Minor changes:

L96: confirmed > contained

L97: gene was confirmed from > genes were detected in

L98: originating from 13 cattle, four pigs, two beef, and three pork, respectively > originating mostly from cattle (13/22), in addition to pigs (4/22) and fresh meat products including beef (2/22) and pork (3/22).

L100-101: Thirteen, three and one E. faecalis isolates were shown to carry optrA and fexA, poxtA and fexA, and optrA, poxtA and fexA, respectively.

L101: has > had

L102: one E. faecium and three E. faecalis harbored poxtA only > two of each E. faecium and E. faecalis isolates harbored poxtA only.

L107-113: Please provide the full form of each antibiotic abbreviation at first use.

L108-109: kanamycin (64%), while the antimicrobial resistance level was low against gentamicin (14%) … and ampicillin (5%).

L111: VAN and SAL-resistant were not identified > no vancomycin or salinomycin resistance was detected

L114: was confirmed to be

Table 2: Please provide the full form of each abbreviation used in the footnote.

L122-123: the MLST analysis of 20 E. faecalis isolates revealed eight sequence types (STs) including ST593 …

L125: whilst ST585

L126:  as > from

L127-128: I would suggest revising these statements to describe that the goeBURST analysis grouped all available STs into six CC and identified ST593, the most prevalent ST type in this study, as a singleton ST, as shown in the minimum spanning tree in Figure 1.

L138: E. faecalis > enterococcal isolates

L140: were > was

Table 4: Please provide further details on the chromosome size, the size range of the plasmid(s), and the number of coding genes in the chromosome and plasmid sequences, respectively.

L154-555: Please make it the starting line of subsection 2.5.1. Also, please consider changing “the mechanisms underlying the antimicrobial-resistant phenotypes” to “the underlying antimicrobial resistance mechanisms”.

L155: Basic Local Alignment Search Tool > blastn search

L169: phenicol resistance genes

L228: characteristics > clonal relationship

L229-230: ST > STs; was variously observed as eight different types > could be classified into eight different types

L240: coharbored

L243: different genes > different combinations of phenicol resistance genes

L259: The optrA gene was in 259 two strains (EFS36 and EFS108) adjacent to transposon Tn554 > The optrA gene in strains EFS36 and EFS108 was located adjacent to Tn554 transposon

L265: for transfer between

L272: flanked

L302: were collected in Korea. Thus, 302 a total of 327 Enterococcus strains were isolated from > were isolated in Korea from

L306: are strains > strains were

L307: identifying > being identified as; France). All enterococci strains were cultured on tryptone …

L320: PCR reaction was carried out with > The PCR positive control DNA

L325: carried > performed

L329: according to

L332: Sixteen antibiotics used in this study were as follows:

L346: were used as the donor cells, and with Escherichia coli strain J53 served as the recipient strain …

Author Response

Response to Reviewer 1 Comments

The manuscript entitled "Prevalence and characteristics of phenicol-oxazolidinone resistance genes in Enterococcus faecalis and Enterococcus faecium isolated from food-producing animals and meat in Korea" is generally well written, with clear presentations. I have only one primary question here and would also like to suggest some minor changes to this work. Please read below for further details.

  1. In E. faecalis strain EFS108, the authors showed that the optrA gene is located on the chromosome, but not in the plasmid. However, the plasmid conjugation experiment demonstrated that this gene has successfully been acquired by the E. coli J53 recipient strain. How did it happen? Can the authors please explain?

Response: We thank you for your critical reviews. This is our mistake. The optrA gene in the E. faecalis strain EFS108 was not acquired by the E. coli J53 recipient strain. Similar to E. faecalis strain EFS108, strain EFS17 showed that the optrA gene is located on the chromosome, and the conjugation experiment demonstrated that this gene has not been acquired by the E. coli J53 recipient strain. The optrA gene loci of the two strains were very similar (Figure 2). We revised the transferability results of the optrA gene in the EFS108 strain in lines 293-294 as follows:

Lines 293-294: optrA and poxtA were detected by broth mating method in 12 and 4 E. faecalis isolates.

Table 1: We revised the transferability of the optrA gene of E. faecalis strain EFS108 from + to -.

Minor changes:

  1. L96: confirmed > contained

Response: As you recommended, we revised the sentence in line 94 as follows:

Line 94: two strains (4.4%) contained the poxtA gene

  1. L97: gene was confirmed from > genes were detected in

Response: As you recommended, we revised the sentence in line 95 as follows:

Line 95: the phenicol-oxazolidinone resistance genes were detected in

  1. L98: originating from 13 cattle, four pigs, two beef, and three pork, respectively > originating mostly from cattle (13/22), in addition to pigs (4/22) and fresh meat products including beef (2/22) and pork (3/22).

Response: As you recommended, we revised the sentence in lines 96-97 as follows:

Lines 96-97: originating mostly from cattle (13/22), in addition to pigs (4/22) and fresh meat products including beef (2/22) and pork (3/22).

  1. L100-101: Thirteen, three and one E. faecalis isolates were shown to carry optrA and fexA, poxtA and fexA, and optrA, poxtA and fexA, respectively.

Response: As you recommended, we revised the sentence in lines 98-100 as follows:

Lines 98-100: Thirteen, four and one E. faecalis isolates were shown to carry optrA and fexA, poxtA and fexA, and optrA, poxtA and fexA, respectively.

  1. L101: has > had

Response: As you recommended, we revised the sentence in line 100 as follows:

Line 100: one E. faecalis had optrA

  1. L102: one E. faecium and three E. faecalis harbored poxtA only > two of each E. faecium and E. faecalis isolates harbored poxtA only.

Response: As you recommended, we revised the sentence in lines 100-101 as follows:

Lines 100-101: two of each E. faecium and E. faecalis isolates harbored poxtA only.

  1. L107-113: Please provide the full form of each antibiotic abbreviation at first use.

Response: As you recommended, we provided the full form of each antibiotic abbreviation in lines 106-110 and 111 as follows:

Lines 106-110: Twenty-two enterococcal strains with phenicol-oxazolidinone resistance genes showed the highest resistance to quinupristin/dalfopristin (100%), florfenicol (100%), tetracycline (95%), erythromycin (95%), tylosin (95%), streptomycin (77%), chloramphenicol (77%), and kanamycin (64%), while the antimicrobial resistance level was low against gentamicin (14%), ciprofloxacin (9%), daptomycin (9%), tigecycline (5%), and ampicillin (5%).

Line 111: Linezolid resistant enterococci

  1. L108-109: kanamycin (64%), while the antimicrobial resistance level was low against gentamicin (14%) … and ampicillin (5%).

Response: As you recommended, we revised the sentence in lines 109-110 as follows:

Lines 109-110: kanamycin (64%), while the antimicrobial resistance level was low against gentamicin (14%), ciprofloxacin (9%), daptomycin (9%), tigecycline (5%), and ampicillin (5%).

  1. L111: VAN and SAL-resistant were not identified > no vancomycin or salinomycin resistance was detected

Response: As you recommended, we revised the sentence in lines 111-112 as follows:

Lines 111-112: no vancomycin or salinomycin resistance was detected

  1. L114: was confirmed to be

Response: As you recommended, we revised the sentence in lines 114-115 as follows:

Lines 114-115: The minimum inhibitory concentration (MIC) of three linezolid-resistant strains was confirmed to be 8 mg/L,

  1. Table 2: Please provide the full form of each abbreviation used in the footnote.

Response: As you recommended, we provided the full form of each abbreviation used in the footnote as follows:

Lines 121-123: 1 AMP, ampicillin; CHL, chloramphenicol; CIP, ciprofloxa-cin; DAP, daptomycin; ERY, erythromycin; FFN, florfenicol; GEN, gentamycin; KAN, kanamycin; LZD, linezolid; STR, streptomycin; TET, tetracycline; TGC, tigecycline; TYLT, tylosin;

  1. L122-123: the MLST analysis of 20 E. faecalis isolates revealed eight sequence types (STs) including ST593 …

Response: As you recommended, we revised the sentence in lines 126-127 as follows:

Lines 126-127: the MLST analysis of 20 E. faecalis isolates revealed eight sequence types (STs) including ST593

  1. L125: whilst ST585

Response: As you recommended, we revised the sentence in line 129 as follows:

Line 129: whilst ST585 (CC4),

  1. L126: as > from

Response: As you recommended, we revised the sentence in line 131 as follows:

Line 131: identified from a strain

  1. L127-128: I would suggest revising these statements to describe that the goeBURST analysis grouped all available STs into six CC and identified ST593, the most prevalent ST type in this study, as a singleton ST, as shown in the minimum spanning tree in Figure 1.

Response: As you recommended, we revised the sentence in lines 131-133 as follows:

Lines 131-133: The goeBURST analysis grouped all available STs into six CC and identified ST593, the most prevalent ST type in this study, as a singleton ST, as shown in the minimum spanning tree in Figure 1.

  1. L138: E. faecalis > enterococcal isolates

Response: As you recommended, we revised the sentence in line 145 as follows:

Line 145: 17 out of 22 (77%) enterococcal isolates.

  1. L140: were > was

Response: As you recommended, we revised the sentence in line 146 as follows:

Line 146: donor strain was successfully delivered

  1. Table 4: Please provide further details on the chromosome size, the size range of the plasmid(s), and the number of coding genes in the chromosome and plasmid sequences, respectively.

Response: As you recommended, we added further details on the chromosome size, the size range of plasmid(s), and the number of coding genes in chromosome and plasmid sequences in Table 4.

Table 4: We added rows for the chromosome size, the size range of plasmid(s), and the number of coding genes in chromosome and plasmid sequences.

  1. L154-555: Please make it the starting line of subsection 2.5.1. Also, please consider changing “the mechanisms underlying the antimicrobial-resistant phenotypes” to “the underlying antimicrobial resistance mechanisms”.

Response: As you recommended, we make the sentence a starting line of subsection 2.5.1. and revised the sentence in lines 150-151 as follows:

Lines 150-151: Complete genome sequencing was performed to identify the underlying antimicrobial resistance mechanisms.

  1. L155: Basic Local Alignment Search Tool > blastn search

Response: As you recommended, we revised the sentence in line 162 as follows:

Line 162: A blastn search for known resistance genes

  1. L169: phenicol resistance genes

Response: As you recommended, we revised the sentence in line 175 as follows:

Line 175: optrA and fexA phenicol resistance genes

  1. L228: characteristics > clonal relationship

Response: As you recommended, we revised the sentence in line 234 as follows:

Line 234: the clonal relationship of phenicol-oxazolidinone

  1. L229-230: ST > STs; was variously observed as eight different types > could be classified into eight different types

Response: As you recommended, we revised the sentence in line 235, 236, and 237 as follows:

Line 235: the STs of enterococci

Line 236: farms could be classified into eight different types

Line 237: the diversity of STs was also reported in enterococci

  1. L240: coharbored

Response: As you recommended, we revised the sentence in line 246 as follows:

Line 246: which coharbored optrA and fexA

  1. L243: different genes > different combinations of phenicol resistance genes

Response: As you recommended, we revised the sentence in line 250 as follows:

Line 250: which have different combinations of phenicol resistance genes,

  1. L259: The optrA gene was in 259 two strains (EFS36 and EFS108) adjacent to transposon Tn554 > The optrA gene in strains EFS36 and EFS108 was located adjacent to Tn554 transposon

Response: As you recommended, we revised the sentence in lines 266-267 as follows:

Lines 266-267: The optrA gene in strains EFS36 and EFS108 was located adjacent to Tn554 transposon

  1. L265: for transfer between

Response: As you recommended, we revised the sentence in line 272 as follows:

Line 272: can be mediated for transfer between different bacterial

  1. L272: flanked

Response: As you recommended, we revised the sentence in line 280 as follows:

Line 280: flanked by IS1216E

  1. L302: were collected in Korea. Thus, 302 a total of 327 Enterococcus strains were isolated from > were isolated in Korea from

Response: As you recommended, we revised the sentence in line 309 as follows:

Line 309: strains were isolated in Korea from 43 meat

  1. L306: are strains > strains were

Response: As you recommended, we revised the sentence in line 313 as follows:

Line 313: 23 E. faecium strains were isolated from

  1. L307: identifying > being identified as; France). All enterococci strains were cultured on tryptone …

Response: As you recommended, we revised the sentence in lines 313-315 as follows:

Lines 313-315: All isolates were used in the experiment after being identified as enterococci using VITEK® MS (bioMérieux, Marcy l’Etoile, France). All enterococci strains were cultured on tryptone soya agar (TSA) medium at 37℃ for 24 h

  1. L320: PCR reaction was carried out with > The PCR positive control DNA

Response: As you recommended, we revised the sentence in line 327 as follows:

Line 327: The PCR positive control DNA for optrA, poxtA, cfr, and fexA

  1. L325: carried > performed

Response: As you recommended, we revised the sentence in line 332 as follows:

Line 332: An antimicrobial susceptibility test was performed on

  1. L329: according to

Response: As you recommended, we revised the sentence in line 336 as follows:

Line 336: according to the manufacturer’s instructions.

  1. L332: Sixteen antibiotics used in this study were as follows:

Response: As you recommended, we revised the sentence in line 339 as follows:

Line 339: Sixteen antibiotics used in this study were as follows

  1. L346: were used as the donor cells, and with Escherichia coli strain J53 served as the recipient strain …

Response: As you recommended, we revised the sentence in lines 353-354 as follows:

Lines 353-354: were used as the donor cells, and with Escherichia coli strain J53 served as the recipient strain

Reviewer 2 Report

The manuscript covers Enterococcus antimicrobial resistance, one of the major public health problems nowadays. The information provided in the manuscript is relevant and demonstrate the alarming problem of antimicrobial resistance in nosocomial bacteria. The authors covers both phenotypic and genotypic information in the manuscript. Did the authors make available the assemblies and raw sequence data is public databases as NCBI? It should be interesting to upload these data and add the accession numbers in the manuscript.

Other comments:

Line 78: Reference for these data?

Line 107: Maybe the authors should highlight that these strains were selected from those with phenicol-oxazolidinone resistance genes.

Line 112-113: I don´t understand the sentence. Except for SYN?

Line 114-115: Please rewrite.

Table 2: Include also the species

Table 3: Please in table footer include the meaning of ST and CC

Line 234: “From this result, resistance was acquired through antibiotics for animals such as florfenicol in the farm”. Could be acquired should be more appropriate. The authors cannot be sure.

Line 292-294: The two sentences seems a bit contradictory.

Line 301: How were the samples analyzed? Only one isolated was selected from each food product? What farms? Poultry, pig…?

Author Response

Response to Reviewer 2 Comments

The manuscript covers Enterococcus antimicrobial resistance, one of the major public health problems nowadays. The information provided in the manuscript is relevant and demonstrate the alarming problem of antimicrobial resistance in nosocomial bacteria. The authors covers both phenotypic and genotypic information in the manuscript. Did the authors make available the assemblies and raw sequence data is public databases as NCBI? It should be interesting to upload these data and add the accession numbers in the manuscript.

Response: We thank you for your critical reviews. As you recommended, we newly uploaded the assemblies for complete genome sequences of E. faecalis EFS17, EFS36, and EFS108. However, we have not yet received accession numbers from NCBI. It may take up to 7 days to receive an accession number from NCBI. We added the submission numbers instead of the accession numbers, which will be replaced by accession numbers later.

Lines 373-375: The complete genome sequences of E. faecalis EFS17, EFS36, and EFS108 were deposited in GenBank under the accession numbers SUB10526591, SUB10526592, and SUB10526593, respectively.

Other comments:

  1. Line 78: Reference for these data?

Response: As you recommended, we added reference for these data in line 78 as follows:

Line 78: roughly twice that of 2010 [12].

  1. Line 107: Maybe the authors should highlight that these strains were selected from those with phenicol-oxazolidinone resistance genes.

Response: As you recommended, we revised the sentence in lines 106-107 as follows:

Lines 106-107: Twenty-two enterococcal strains with phenicol-oxazolidinone resistance genes showed the highest resistance to quinupristin/dalfopristin

  1. Line 112-113: I don´t understand the sentence. Except for SYN?

Response: This sentence was deleted as it is irrelevant to the results.

  1. Line 114-115: Please rewrite.

Response: As you recommended, we revised the sentence in lines 114-116 as follows:

Lines 114-116: The minimum inhibitory concentration (MIC) of three linezolid-resistant strains was confirmed to be 8 mg/L, and the MIC of five linezolid-intermediate strains was determined to be 4 mg/L (Table 2).

  1. Table 2: Include also the species

Response: As you recommended, we included the species in Table 2.

Table 2: We newly added a column for species.

  1. Table 3: Please in table footer include the meaning of ST and CC

Response: As you recommended, we added the meaning of ST and CC in table footer in line 135 as follows:

Line 135: 1 ST, sequence type; 2 CC, clonal complex.

  1. Line 234: “From this result, resistance was acquired through antibiotics for animals such as florfenicol in the farm”. Could be acquired should be more appropriate. The authors cannot be sure.

Response: As you recommended, we revised the sentence in line 240 as follows:

Line 240: From this result, resistance could be acquired through antibiotics for animals such as florfenicol in the farm

  1. Line 292-294: The two sentences seems a bit contradictory.

Response: As you recommended, we revised the sentence in lines 299-301 as follows:

Lines 299-301: In conclusion, although the occurrence of phenicol-oxazolidinone resistance gene in enterococci is still rare among food animals, a high rate of transferable phenicol-oxazolidinone genes was observed in these strains.

  1. Line 301: How were the samples analyzed? Only one isolated was selected from each food product? What farms? Poultry, pig…?

Response: As you recommended, we added the sentence for number of samples analyzed in lines 309-310 as follows:

Lines 309-310: strains were isolated in Korea from 43 meat (19 beef and 24 pork), 24 slaughterhouses, and 16 farms.

Reviewer 3 Report

Manuscript ID: ijms-1415638

Review Report

This is a report for the manuscript entitled “Prevalence and characteristics of phenicol oxazolidinone resistance genes in Enterococcus faecalis and Enterococcus faecium isolated from
food-producing animals and meat in Korea”.

This is a manuscript with adequate and comprehensive data supported by adequate tables and figures, which contributes to increase the knowledge about the prevalence of resistance genes in bacteria of animal origin with the potential to be transmitted to humans through food chains.

The authors analyzed the prevalence and characteristics of phenicol-oxazolidinone resistance genes in many isolates of two species of Enterococcus (E. faecalis and E. faecium) and reported, for the first time, the genome sequence of three E. faecalis isolates possessing phenicol-oxazolidinone resistance genes both in the chromosome and in a plasmid. They also found several mobile gene elements and transposase-associated genes, enabling the horizontal transfer of the phenicol-oxazolidinone resistance and other antimicrobial-resistant genes.

Although the study is relevant and the results appear reliable, the manuscript needs to be rewritten in some parts, since sometimes the meaning of the sentence does not fully correspond to what is being demonstrated. Another recurrent issue that needs to be fixed is the appearance of abbreviations without definition.

Specific comments:

Abstract

Abbreviations needing definition, for example in lines 21, 24, 25, 26.

Introduction

Abbreviations needing definition.

The last sentence (lines 87 – 89) seems to be more a summary than one objective of the study.

Results

Lines 122 – 127: Sentence that needs to be re-written and abbreviations must be defined.

Table 3 (Line 129): ST and CC only defined in the legend of Figure 1!

Lines 148, 150, 151, and Table 4: RNAs are not present in genomes. The authors should refer to “coding regions for RNAs”.

Line 166: Please provide the meaning of ISs.

Figure 5 (Line 185): This is not a clear image. The font size should be increased.

Materials and Methods

Line 314 – 316: Example of a sentence needing complete revision.

Lines 332 – 336: Sentence needing revision as well as its meaning.

Author Response

Specific comments:

Abstract

  1. Abbreviations needing definition, for example in lines 21, 24, 25, 26.

Response: We thank you for your critical reviews. optrA, poxtA, fexA, IS1216E, and Tn554 are gene names and was used as it is without definition in other studies (Li et al., 2019, doi: 10.1128/AAC.00809-19; Shan et al., 2020, doi: 10.1093/jac/dkaa325; Wang et al., 2015, doi: 10.1093/jac/dkv116). Also, EFS17, EFS36, and EFS108 are strain name, not an abbreviation. As you recommended, we added definition in line 22 as follows:

Line 22: sequence types (STs)

Introduction

  1. Abbreviations needing definition.

Response: As you recommended, we added definitions for abbreviations in lines 49 and 52 as follows:

Line 49: ribosomal ribonucleic acid (rRNA)

Line 52: adenosine triphosphate

  1. The last sentence (lines 87 – 89) seems to be more a summary than one objective of the study.

Response: As you recommended, we deleted the last sentence as it is irrelevant to the objective.

Results

  1. Lines 122 – 127: Sentence that needs to be re-written and abbreviations must be defined.

Response: As you recommended, we revised the sentence and added definitions for abbreviations in lines 126-133 as follows:

Lines 126-133: As shown in Table 3, the MLST analysis of 20 E. faecalis isolates revealed eight sequence types (STs) including ST593, ST100, ST16, ST585, ST915, ST338, ST47, and ST27. ST593 (clonal complex (CC) singleton) was the most predominant with nine strains (43%), and ST100 (CC100) and ST16 (CC16) were determined from three strains each, whilst ST585 (CC4), ST 915 (CC915), ST338 (CC4), ST47 (CC47), and ST27 (CC27) were each identified from a strain. The goeBURST analysis grouped all available STs into six CC and identified ST593, the most prevalent ST type in this study, as a singleton ST, as shown in the minimum spanning tree in Figure 1.

  1. Table 3 (Line 129): ST and CC only defined in the legend of Figure 1!

Response: As you recommended, we added the meaning of ST and CC in table footer in line 135 as follows:

Line 135: 1 ST, sequence type; 2 CC, clonal complex.

  1. Lines 148, 150, 151, and Table 4: RNAs are not present in genomes. The authors should refer to “coding regions for RNAs”.

Response: As you recommended, we revised the sentence in lines 156, 157, and 158-159 as follows:

Line 156: 73 coding regions for RNAs

Line 157: 78 coding regions for RNAs

Lines 158-159: 73 coding regions for RNAs

  1. Line 166: Please provide the meaning of ISs.

Response: As you recommended, we added the meaning of ISs in line 172 as follows:

Line 172: insertion sequences (ISs),

  1. Figure 5 (Line 185): This is not a clear image. The font size should be increased.

Response: As you recommended, we revised the clear image (300 dpi) of Figure 3 and increased the font size.

Materials and Methods

  1. Line 314 – 316: Example of a sentence needing complete revision.

Response: As you recommended, we revised the sentence in lines 321-323 as follows:

Lines 321-323: The PCR mixture (25 µl) contained 1 µl of each primer (0.4 µM), 0.1 µl of Taq polymerase (5U/µl), 2.5 µl of 10× buffer, 16.4 µl of distilled water, and 2 µl of template DNA.

  1. Lines 332 – 336: Sentence needing revision as well as its meaning.

Response: As you recommended, we revised the sentence in lines 339-343 as follows:

Lines 339-343: Sixteen antibiotics used in this study were as follows; gentamycin (GEN), streptomycin (STR), kanamycin (KAM), ampicillin (AMP), ciprofloxacin (CIP), vancomycin (VAN), tigecycline (TGC), erythromycin (ERY), tylosin (TYLT), linezolid (LZD), chloramphenicol (CHL), florfenicol (FFN), quinupristin/dalfopristin (SYN), tetracycline (TET), daptomycin (DAP), salinomycin (SAL).